# Research on Online Sales of Low-Carbon Agricultural Products by New Urban Agricultural Business Entities: Evidence from Shanghai, China

**Fan Xu** [1] , **Fangke Xu** [1], **Pu Xu** [1] , **Min Liu** [2,*] **and Yufeng Li** [1]

[1] School of Economics and Management, Shanghai Ocean University, Shanghai 201306, China; xufan0922@163.com (F.X.); 2143118@st.shou.edu.cn (F.X.); pxu@shou.edu.cn (P.X.); liyf@shou.edu.cn (Y.L.)
[2] School of Humanities and Management, Guilin Medical University, Guilin 541199, China
\* Correspondence: liumin@glmc.edu.cn

**Abstract:** Traditional agricultural business entities present environmental problems such as high energy consumption and high pollution. To achieve goals such as carbon capping and carbon neutrality, governments should encourage urban agricultural operators to sell low-carbon agricultural products online. This could play an important role in smoothing the connection between production and marketing, achieving industrial prosperity and promoting low-carbon agricultural development. This paper explores the formation and behavioral transformation of online sales intention by combining the theory of planned behavior, the Motivation–Opportunity–Ability (MOA) model, and the binary Probit regression model using data from 106 questionnaires. The study found that economic expectations and social norms can significantly improve online sales intention during the formation stage of online sales intention. Behavior control is not conducive to improving online sales intention. In addition, we found a gap between the willingness of urban agricultural operators to sell online and their behavior. This gap requires activation and adjustment of the opportunity and capability factors in the behavior transformation phase. Finally, we found that the strengthening of policy conditions and management capacity facilitated the transformation of urban agricultural operators' willingness to sell online into behavior. This paper provides recommendations for the online sales of low-carbon agricultural products. While we continue to deepen urban agricultural operators' knowledge of online sales, we should also pay attention to the creation of external opportunities that are suitable for the development of online sales, and identify differences in capacity among operators. This will provide precise services.

**Keywords:** new urban agricultural business entities; low-carbon agricultural products; online sales; willingness and behavior; theory of planned behavior; MOA model





## 1. Introduction

Global warming presents serious challenges to human survival and development, and agriculture is highly vulnerable to climate change due to its natural production properties [1]. As an important industrial sector, the low-carbon development of agriculture is increasingly attracting the attention of the public, and low-carbon agriculture has become the future development direction of agriculture. However, the development of low-carbon agriculture in China still faces problems such as the prominent contradiction between man and land, the fragile agricultural ecological environment, and insufficient understanding of low-carbon agriculture among farmers [2]. The operation mode consisting of individual small farmers is unable to meet the development needs of low-carbon agriculture. However, urban agricultural business entities are regarded as an important force in the development of low-carbon agriculture due to their high level of education and professional production. They can realize the unity of economic, social, and ecological benefits by producing and selling low-carbon agricultural products [3].

Low-carbon agricultural products also have the biological characteristics of being perishable and difficult to keep fresh. Moreover, the production of agricultural products is regional and seasonal. This determines the fact that farmers have less time and must perform difficult tasks during the sales process. The rapid development of online sales can provide new sales channels for low-carbon agricultural products. The expansion of sales channels can expand the sales radius, reduce circulation links, and improve circulation efficiency. This will further promote farmers' income and development. The No. 1 document issued by the Central Committee of the Communist Party of China and the State Council in 2023 also pointed out that it is necessary to implement the "Prosperity of Agriculture through Business" and "Internet +" projects in detail in order to achieve the export of agricultural products from villages to cities. However, agricultural products must go through multiple steps between production and final sale, such as seed selection, seedling breeding, fertilizer and pesticide application, etc. Any problems at any step will affect the product perception among consumers. The high degree of virtuality of online sales exacerbates the information asymmetry in the agricultural products market. In this case, the carbon label of low-carbon agricultural products can effectively convey product quality information and connect consumers with agricultural business entities.

Carbon labeling refers to the quantified index of greenhouse gas emissions throughout the whole life cycle of a product, namely, the whole process from raw material, through manufacturing, storage, and transportation, to waste and recycling [4]. This unique mark of product quality and service can provide benefits to agricultural operators and convey clear and credible information to consumers. Therefore, carbon labels could become an important tool allowing agricultural operators to seize the market. However, the exploration of carbon labeling and low-carbon products in China is still in its infancy [5]. The development of low-carbon agricultural product systems still faces institutional and technical constraints; therefore, this paper uses the "three products and one standard" quality certification of agricultural products to represent low-carbon agricultural products. The price of low-priced agricultural products is usually higher than that of ordinary agricultural products. Ordinary agricultural products have obvious price advantages and the advantage of premium advertising. However, consumers need to upgrade their consumption concept and pay attention to low-carbon agriculture and the low-carbon economy.

Domestic and international scholars have mostly used the definition of carbon labels given by the UK Carbon Trust as the standard, whereby carbon labels are a new type of eco-label identifying products or services in terms of a quantitative index, providing consumers with information regarding the total amount of carbon dioxide produced throughout the entire life cycle of the product or service [6]. With carbon labels attached to agricultural products, consumers have access to information on the carbon emissions of agricultural products. This will expand the public's awareness of agricultural products with carbon labels, thus improving the visibility of agricultural products as well as public participation, guiding consumers to buy low-carbon goods [7]. In terms of the impact of low-carbon agricultural products on online sales, China has not yet developed a systematic carbon labeling system due to the country's late start in the field of low-carbon agricultural products. Combined with the current development situation, agricultural products with "three products and one standard" certification will treated as representative of low-carbon agricultural products. "Three products and one standard" is a public brand denoting safe, high-quality agricultural products instituted by the government. This meets consumers' demands with respect to the quality of agricultural products, and guarantees the products' quality and safety. The existing literature shows that agricultural products with digital labels can reduce consumers' concerns about product quality and safety and brand credibility, force the production end to strengthen the management of agricultural products, and alleviate information asymmetry in online sales [8,9]. For new agricultural business entities, marked agricultural products can improve operators' agricultural income and promote the growth of the agricultural economy [10]. Regardless of whether agricultural products possess

"three products and one standard" certification, or are marked with the operator's personal trademark, these all have a significant positive impact on online sales [11].

When it comes to the sale of agricultural products online, scholars have mostly focused on small farmers. Cui BD et al., Lin HY et al. investigated the effect of personality factors on readiness to sell online. Farmers who were younger and more educated were more likely to engage in agricultural product e-commerce [12,13]. In terms of business characteristics, according to Zhang YF, characteristics influencing farmers' willingness to sell agricultural products online included the scale of their planting operations, their income, and whether they participated in cooperatives [14]. The factors influencing online sales behavior overlap with those that influence e-commerce willingness. Zeng YW et al. claimed that individual and business factors had a significant impact on farmers' e-commerce behavior with respect to agricultural products [15]. Behavior studies have been performed focusing on e-commerce technologies and external environmental factors. With respect to the strengthening of online sales skills, Yan BB et al. discovered that farmers' information literacy had a large positive impact on their online sales behavior [16]. According to Mu YH, Han J et al., in the external environment, infrastructure and government assistance are key driving elements for farmers' online sales behavior [17,18].

Research has been performed on the manner in which online sales willingness is translated into behavior. Some scholars believe that willingness can accurately predict behavior: if the willingness is stronger, the behavior will be more likely to occur [19]. Meanwhile, some scholars believe that even if there is a high level of willingness, there will be a large deviation between willingness and behavior. Xiong X and Nie FY found, based on the mean T-test method, that among farmers with online sales willingness, whether this translates into behavior will be affected by infrastructure and the operators' e-commerce skills [20]. Wang XD also reported that there was a large deviation between farmers' online sales intention and their behavior. This was a result of factors such as personal characteristics, policy perception, and degree of understanding of network information [21]. However, they did not analyze the factors affecting the transformation of online sales willingness for low-carbon agricultural products into actual behavior.

In summary, there is an abundance of studies on low-carbon agricultural products and online sales of agricultural products in the existing literature. However, current research still has the following deficiencies. First, the existing research is focused more on small farmers in traditional agricultural areas, and pays less attention to urban areas. The development of urban agriculture cannot be ignored. Second, agricultural products with "three products and one standard" quality certification are regarded as being representative of China's low-carbon agricultural products. Few studies have been performed addressing the ways in which urban agricultural business entities drive the development of low-carbon agriculture by selling low-carbon agricultural products. Third, few studies have been performed that include online sales willingness and behavior at the same time, and there is a lack of research on the factors restricting or promoting the transformation of willingness into behavior.

In this paper, new agricultural business entities in Shanghai are selected as the research object. First, as a megacity, Shanghai has a complete infrastructure, including transportation and communication networks, as well as other resources; it also has the advantage of supporting the online sale of low-carbon agricultural products. Secondly, according to the latest statistics, the per capita income of Shanghai was close to CNY 80,000 in 2022, ranking first in China. Consumers in Shanghai have higher requirements in terms of the quality of agricultural products, and have access to a broad market for the sale of low-carbon agricultural products. Finally, in the post-epidemic era, the disruption and disorder of the urban food chain have gradually brought attention to the function of Shanghai in agricultural production [22]. Compared with ordinary agricultural areas, agriculture in Shanghai presents problems such as a shortage of agricultural land, high agricultural labor costs, and a shortage of rural labor. It is therefore necessary to develop agricultural products with high added value. Low-carbon agricultural products have price advantages

as high-quality agricultural products, and the development of online sales for low-carbon agricultural products is an important strategy for Shanghai in promoting high-quality agriculture. On the basis of field research and the questionnaire, the research group learned that most agricultural managers in Shanghai are willing to sell low-carbon agricultural products online. However, the actual participation rate in online sales of low-carbon agricultural products is low in Shanghai due to factors such as the limited knowledge of e-commerce operators, the imperfection of existing logistics systems, and the high cost of developing new sales channels. Against the backdrop of achieving the goal of "double carbon", it is necessary to explore the factors affecting the willingness of Shanghai's new agricultural business entities to sell low-carbon agricultural products online. This is not only conducive to improving the level of development of low-carbon agriculture. It can also provide consumers with high-quality low-carbon agricultural products, thereby promoting the development of low-carbon agriculture.

In summary, the transformation process of the willingness of firms in urban areas to pursue online sales of low-carbon agricultural products into behavior in this regard, and its main influencing factors, is unclear, and we will attempt to fill this gap. In this paper, the theory of planned behavior is combined with the MOA model, and new agricultural management entities in Shanghai are taken as the research object. In this paper, agricultural products with "three products and one standard" quality certification are taken to represent low-carbon agricultural products, and motivational factors are taken as the independent variables affecting the willingness of new agricultural management subjects to sell low-carbon agricultural products online. The results of this study reveal the role played by opportunity and capability factors in the path from willingness to behavior. In addition, the findings provide constructive suggestions for the government in promoting the development of low-carbon agriculture and online sales.

## 2. Theoretical Framework

According to the theory of planned behavior, behavior is often impacted by behavioral attitudes, arbitrary standards, and perceived behavioral control. This is a critical conceptual foundation for understanding and investigating individual behaviors [23]. This theory focuses on the study of individual subjective cognition and willingness as an important pre-variable determining whether behavior will occur, in light of current behavioral research showing that individual willingness cannot always predict behavior, and that a person's subjective cognition only has a small impact on behavioral decision making [24]. Therefore, the academic community has proposed the MOA model. The MOA model was first used to study individual information processing behavior. To start with, only motivation and ability factors were considered to have an impact on personal information processing, but this was then expanded to three variables: motivation, opportunity, and ability [25]. In previous studies, motivation has been regarded as the motivation to achieve the goal, while opportunity and ability affect whether the individual behavior is able to occur. This theory focuses on the discussion of the external environment and objective ability. Objective constraints, can improve the explanatory power of decision making with respect to individual behavior. Planned behavior theory has been introduced into the MOA model, with the attitude, subjective norms and perceived behavior control of planned behavior theory being adopted as the overall motivation influencing individual behavioral intention and behavior in the MOA model [26]. Only when an individual has both opportunity and ability elements will an individual's willingness be transformed into behavior.

This paper defines the online sale of low-carbon agricultural products by new agricultural business entities as the use of Internet technology. Enterprises deliver low-carbon agricultural products directly to consumers through online transactions and logistics distribution. In recent years, the development of the online sale of low-carbon agricultural products has been accompanied by government support and leadership. This has prompted most operators to have a strong willingness to sell online. On the basis of thinking as a "rational person", new agricultural business entities should adopt online sales behavior,

but when faced with reality, entities' online sales behavior with respect to low-carbon agricultural products is often lower than online sales willingness. The analysis of factors influencing business entities' willingness to sell low-carbon agricultural products online is insufficient to explain the response in the form of business entities' online sales behavior. In addition to the willingness to control, more conditions are required in business entities' transformation process towards online sales behavior. Therefore, by combining the subjective cognitive perspective in accordance with the Theory of Planned Behavior (TPB) and the objective constraints perspective in accordance with the MOA model, this paper takes the motivation of new agricultural operators as the key factor affecting the generation of willingness and behavior. We examine the role of opportunity and ability factors in the transformation of online willingness into behavior among new agricultural operators.

### 2.1. Motivation Affects the Willingness and Behavior of New Agricultural Business Entities to Sell Low-Carbon Agricultural Products Online

According to Westaby, individual behavioral willingness is a direct function of overall motivation, whereby he identified attitude, subjective norm, and perceived behavior control as the three main drivers of overall motivation [27]. One of these is attitude, which is a person's positive or negative opinion of a specific activity. If an agricultural business entity has a positive impression of online sales, it will have a positive attitude [23]. According to the rational farmer theory, under completely rational conditions, peasant households will make production and management decisions in order to maximize profits. If the agricultural business entity expects better economic outcomes to follow the implementation of online sales of low-carbon agricultural products, it will have a positive attitude and evaluation that this behavior will help enhance income, expand sales, and meet consumer demand. Subjective norms are the pressures that people feel from social institutions or other people. They usually contain both prescriptive norms and exemplary norms [28]. These norms usually refer to social norms. An entity's behavior regarding online sales can be influenced by the people and environment around them. Because there are no prescriptive rules for establishing online sales channels, this paper focuses on exemplary standards and explores the ways in which entities base their decisions on the opinions of their friends, family, and neighbors, as well as how other entities are run. The desire of other operators not to be left behind will increase the willingness and behavior of other commercial entitiesvis a vis participation in online sales, especially once those pioneers in the field have seen certain benefits. Perceived behavior control refers to people's subjective ability to carry out a specific behavior. To avoid misunderstandings with the opportunity and ability parts of the MOA model, this study exclusively analyzes the perceived difficulty and perceived control of online sales technology in order to gauge the control of business entities over online sales willingness and behavior. On this basis, the following research hypotheses are proposed.

**H1a:** *The greater the expected economic benefits of online sales, the more willing urban agricultural business entities will be to sell low-carbon agricultural products online.*

**H1b:** *The stronger the social norms function of online sales, the more willing urban agricultural business entities will be to sell low-carbon agricultural products online.*

**H1c:** *The easier it is to control online sales, the more willing urban agricultural business entities will be to sell low-carbon agricultural products online.*

This paper presents three research hypotheses (H2a, H2b, and H2c). The motivations described above will continue to influence the online selling behavior of urban agricultural business entities. We therefore propose the following hypotheses.

**H2a:** *The greater the expected economic benefits of online sales, the more likely urban agricultural operators will be to sell low-carbon agricultural products online.*

**H2b:** *The stronger the social norms function of online sales, the more likely urban agricultural operators will be to sell low-carbon agricultural products online.*

**H2c:** *The easier it is to control online sales, the more likely urban agricultural operators will be to sell low-carbon agricultural products online.*

*2.2. Opportunity to Influence the Transformation of Urban Agricultural Business Entities' Willingness to Sell Low-Carbon Agricultural Products Online into Behavior*

Opportunities are valuable factors in the objective, outside environment that might influence a person's behavior. The natural environment and the social environment are two distinct sorts of context in which humans may produce, store, or transmit knowledge [29]. In this paper, by combining the characteristics of low-carbon agricultural products and the characteristics of online sales behavior, the effective ingredients that are conducive to stimulating urban agricultural business entities to carry out online sales are selected. Natural conditions have a significant impact on and constrain agricultural production in accordance with the natural environment [30]. As farmland size and terrain quality improve, production costs will fall, and entities' ability to develop new technologies grows. Only in the presence of natural conditions can online sales willingness be effectively transformed into behaviors. In terms of the social environment, the government's active policies can reduce the cost incurred by businesses when establishing online sales and increase the possible benefits to businesses. If businesses benefit immediately from online sales intent, they have an increased chances of converting this intent into action. Based on this, the following research hypothesis is proposed:

**H3:** *Opportunity has a significant moderating effect on the transformation of urban agricultural business entities' willingness to sell low-carbon agricultural products online into behaviors.*

*2.3. Ability to Influence the Transformation of Urban Agricultural Business Entities' Willingness to Sell Low-Carbon Agricultural Products Online into Behavior*

Ability can be classified into three types, depending on function: cognitive ability, operational ability, and social ability. Ability is defined as an individual's intrinsic potential, and determines the quality and quantity of activities completed [29]. After a business entity has formed the willingness to sell online, it will evaluate its own ability to participate, and then generate online sales behavior. Operators must dedicate human, material, and financial resources to expanding online sales channels. Ability is divided into two categories in this report: fundamental abilities and technological abilities [31]. Level of education is seen as a fundamental ability, whereas agricultural income is regarded as an economic ability. The level of mastery of online sales technology of agricultural business entities is introduced as a technical ability in order to determine the ability factors fully and exactly. The income level of the operator provides the material security required for the expansion of online sales channels, and the operator's level of education and competency with internet technology can help them conduct online sales. In order to successfully convert willingness into behavior, operators must have the ability, in terms of both software and hardware, to successfully engage in online sales from their initial desire to do so through to the occurrence of the final action. Based on this, the following research hypothesis is proposed:

**H4:** *Ability has a significant moderating effect on the transformation of urban agricultural business entities' willingness to sell low-carbon agricultural products online into behavior.*

Figure 1 displays the theoretical analytical framework of this paper based on the research assumptions outlined above.

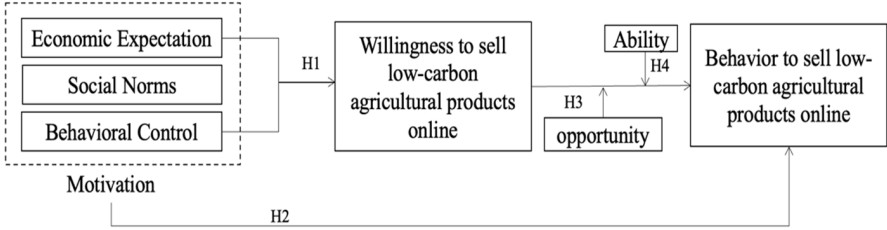

**Figure 1.** Theoretical analysis framework.

### 3. Materials and Methods

#### 3.1. Materials

This study takes the basic components of agricultural management in Shanghai as its research object. Shanghai has a dense transportation network. Good transportation capacity ensures the supply of agricultural products. Shanghai has advantages in terms of policy. Shanghai has introduced many agricultural support policies that are conducive to the production and online sales of low-carbon agricultural products. Secondly, Shanghai's new agricultural business entities are cultivating remarkable results. Family farms, farmers' cooperatives, and leading agricultural enterprises, as the main agricultural business entities, are developing rapidly and taking responsibility for joint farming with farmers. Thirdly, agricultural products from Shanghai have a prominent brand advantage, increasing the added value of agricultural products. Through brand building, the quality of agricultural products can be signaled to consumers, and consumers' awareness of the quality of low-carbon agricultural products can be improved, thus promoting the ability of new agricultural business entities to obtain more online traffic and improving their earnings.

The research team spent more than 30 h visiting the Shanghai Municipal Agricultural and Rural Committee, Shanghai Pudong New Area Agricultural Association, and 15 agricultural business entities between 8 January and 18 January, 2022. To overcome sample bias, we strictly standardized the survey process. We produced interview transcripts amounting to about 100,000 words in nine agricultural parks in Shanghai. The purpose of this was to achieve an in-depth understanding of willingness and actual behavior with respect to the online sale of low-carbon agricultural products. To avoid information bias, we introduced the basic concept of the online sale of green agricultural products to the enterprises before the survey. We distributed 112 questionnaires to people in charge of agricultural business entities. The questionnaire was centered on online sales of agricultural products. This mainly included willingness to sell online, understanding of online sales, problems with online sales, and possible future sales methods. With respect to the interview content and supporting documentation, 106 valid responses to the questionnaire were recovered, corresponding to a recovery percentage of 94.64%.

#### 3.2. Methods

The explained variables were the willingness and behavior of business entities to sell low-carbon agricultural products online. Values were set to be equal to "0" or "1" to indicate the value of the observed willingness and behavior, where "0" indicates that the operating entity is willing to sell wirelessly or exhibits wireless sales behavior, while "1" indicates that the operating entity is willing to sell online or exhibits online sales behavior. As previously stated, these are discrete binary variables. As a result, the binary probit model was used to account for the impact of various motivational components on business entities' willingness and behavior to engage in online sales. Formula (1) presents the expressions for models 1 and 2, where $y_i$ ($i$ = 1, 2) denotes the previously explained variables; $x_i$ denotes an explanatory variable that influences the business entity's willingness to engage in online sales; $\beta_0$ denotes a constant term; and $\beta_i$ denotes the regression coefficient of the explanatory variable.

$$p(y_i = 1 | X_i) = F(X_i, \beta) = F(\beta_0 + \beta_1 X_1 + \beta_2 X_2 + \ldots + \beta_n X_n) \tag{1}$$

The moderating effects of the opportunity and ability factors on willingness and behavior to sell low-carbon agricultural products online were examined. With reference to the moderation effect test method proposed in [32], an interaction term analysis was performed in which the moderating variable, opportunity, was a continuous variable and the online sales behavior was a categorical variable. The $y$ in Formula (2) represents the corporate entity's online sales activity. If $y = 1$, the business entity engages in online sales activity; otherwise, it does not. The willingness of the business entity to engage in online sales is represented by $x$, $z_i$ represents moderating variables (natural conditions, policy conditions), $m$ represents other control variables, and $\beta_i$ represents the regression coefficient of the explanatory variable.

$$p(y = 1|X) = p(y = 1|X, Z_i, m) = F(\beta_0 + \beta_0 X + \beta_1 Z_i + \beta_2 X Z_i + \beta_2 m) \tag{2}$$

When the ability of the moderator variable is categorical, it is separated into two groups, one with high ability and the other with low ability, on the basis of the mean value. The operator's ability is measured by comparing the difference between the regression coefficients of the two groups of samples. Grouping regression is used to verify and examine the two groups of samples in terms of whether or not there is a regulatory effect.

### 3.3. Variable Selection

Based on the theoretical analytical framework described above, two indicators are applied in this study to identify the explanatory variables: "whether the entity is willing to sell low-carbon agricultural products online", and "whether is the entity participates in behavior to sell low-carbon agricultural products online". The primary explanatory component is manager motivation. The core explanatory variable is operator motivation. Combined with the theory of planned behavior and the research focus of this paper, the behavioral attitude of business subjects is expressed on the basis of their expectations regardin the economic benefits they can obtain from online sales. Subjective norms mainly come from the social pressure felt by business entities to participate in online sales. This pressure mainly comes from reference groups around them, such as family members, friends, and agricultural technicians. Perceived behavior control is derived from the potential difficulties that business subjects perceive may arise in online sales. Therefore, a total of eight items were selected from the three dimensions of economic expectation, social norms, and behavior control, all of which were selected from existing mature scales, as can be observed from previous studies [33,34]. The MOA model was combined with the research focus of this paper in consideration of the fact that opportunity and ability can transform online sales intention into behavior, and natural conditions and policy conditions were selected to represent opportunity factors [29]. The level of education, income level, and Internet technology level of the operators were selected to represent the ability factors [30]. The control variables were the organizational and personal characteristics of the operator, including the type of business subject, gender, age, and non-agricultural income. Table 1 gives full descriptions of and descriptive statistics for each variable.

### 3.4. Basic Characteristics of the Sample

The basic characteristics of the valid samples are shown in Table 2. Since different types of new agricultural business entity can be operating concurrently, the samples are counted at the highest level of the business entity. Regarding the business entity types, more than 60% (61.32%) of the entities were farmer cooperatives; the cultivated land area of the entities was concentrated between 6.67 and 33.3 hectares, accounting for 58.49% of the sample; more than 60% of the land of the main business was reported to be contiguous. From the perspective of gender, the ratio of males and females was relatively balanced, with males accounting for 59.44% and females accounting for 40.56%; the age of the respondents was concentrated between 31 and 50 years of age, accounting for 74.53%; the level of education was generally high, with the proportion of respondents holding a college degree or above

being 75.47%. In terms of what proportion of household income was non-agricultural, the sample was more consistent, and was mostly higher than 40%.

**Table 1.** Variable definitions and descriptive statistics.

| Variable Type | Variable Name | Mean | Standard Deviation |
|---|---|---|---|
| Explained variable | Willingness to sell low-carbon agricultural products online no = 0, yes = 1 | 0.698 | 0.461 |
| | Behavior of selling low-carbon agricultural products online: no = 0, yes = 1 | 0.509 | 0.502 |
| Core explanatory variable | Economic Expectation (EE) | | |
| | Enhance income (EE1): Very low = 1; Low = 2; Normal = 3; High = 4; Very high = 5 | 3.349 | 1.042 |
| | Expand sales (EE2): Very low = 1; Low = 2; Normal = 3; High = 4; Very high = 5 | 3.424 | 0.925 |
| | Meet consumer demand (EE3): Very low = 1; Low = 2; Normal = 3; High = 4; Very high = 5 | 3.330 | 0.992 |
| | Social Norms (SN) | | |
| | Influenced by relatives and friends (SN1): Very low = 1; Low = 2; Normal = 3; High = 4; Very high = 5 | 3.283 | 1.021 |
| | Influenced by other business entities (SN2): Very low = 1; Low = 2; Normal = 3; High = 4; Very high = 5 | 3.292 | 1.059 |
| | Influenced by agricultural technicians (SN3): Very low = 1; Low = 2; Normal = 3; High = 4; Very high = 5 | 3.792 | 0.963 |
| | Behavioral Control (BC) | | |
| | Difficulty of participating in online sales (BC1): Very low = 1; Low = 2; Normal = 3; High = 4; Very high = 5 | 3.106 | 1.043 |
| | Difficulty of operating online sales (BC2): Very low = 1; Low = 2; Normal = 3; High = 4; Very high = 5 | 3.566 | 0.995 |
| Moderator | Opportunity | | |
| | Cultivated area: 0–6.67 hectares = 1; 6.67–20 hectares = 2; 20–33.33 hectares = 3; 33.33–66.67 hectares = 4; over 66.67 hectares = 5 | 2.632 | 1.221 |
| | Whether the land is contiguous: no = 0, yes = 1 | 0.651 | 0.479 |
| | Government propaganda: Very low = 1; Low = 2; Normal = 3; High = 4; Very high = 5 | 3.377 | 1.125 |
| | Government subsidy: Very low = 1; Low = 2; Normal = 3; High = 4; Very high = 5 | 3.321 | 0.991 |
| | Ability | | |
| | Education: Junior high school and below = 1; Technical secondary school or high school = 2; Junior college = 3; Undergraduate and above = 4 | 3.047 | 0.939 |
| | Farm income: Under CNY 50,000 = 1; CNY 50,000–10,000 = 2; CNY 10,000–30,000 = 3; CNY 30,000–80,000 = 4; Over CNY 80,000 = 5 | 3.896 | 1.145 |
| | Proficiency at operating electronic equipment: Very low = 1; Low = 2; Normal = 3; High = 4; Very high = 5 | 2.754 | 0.903 |
| | Frequently browse agricultural product information online: Very low = 1; Low = 2; Normal = 3; High = 4; Very high = 5 | 3.462 | 1.052 |
| Control variable | Business entity scale: Family farm = 1; Professional cooperative = 2; Big enterprise = 3 | 1.745 | 0.570 |
| | Sex: Female = 0; Male = 1 | 0.594 | 0.493 |
| | Age: Under 30 years old = 1; 30–40 years old = 2; 40–50 years old = 3; 50–60 years old = 4; Over 60 years old = 5 | 3.471 | 0.841 |
| | Proportion of non-agricultural income: Below 10% = 1; 10–40% = 2; 40–60% = 3; 60–90% = 4; above 90% = 5 | 3.009 | 1.116 |

**Table 2.** Statistical results regarding the basic characteristics of the sample.

| Basic Characteristic | Variable Category | Proportion | Basic Characteristic | Variable Category | Proportion |
|---|---|---|---|---|---|
| Type of business entity | Family farm | 32.07% | | 0–6.67 hectares | 16.98% |
| | Professional cooperative | 61.32% | Cultivated area | 6.67–20 hectares | 37.74% |
| | Big enterprise | 6.61% | | 20–33.33 hectares | 20.75% |
| Sex | Male | 59.44% | | 33.33–66.67 hectares | 14.15% |
| | Female | 40.56% | | Over 66.67 hectares | 10.38% |
| | Under 30 years old | 4.72% | | Junior high school and below | 8.49% |
| Age | 31–40 years old | 38.68% | Education | Technical secondary school or high school | 16.04% |
| | 41–50 years old | 35.85% | | Junior college | 37.74% |
| | 51–60 years old | 17.92% | | Undergraduate and above | 37.73% |
| | Over 61 years | 2.83% | | Under CNY 50,000 | 3.77% |
| | Under 10% | 7.55% | | CNY 50,000–10,000 | 6.60% |
| Non-agricultural income as a share of household income | 10–40% | 26.42% | Agriculture income | CNY 10,000–30,000 | 28.30% |
| | 40–60% | 35.85% | | CNY 30,000–80,000 | 18.87% |
| | 60–90% | 17.92% | | Over CNY 80,000 | 42.46% |
| | Over 90% | 12.26% | Whether the land is contiguous | Yes | 65.09% |
| | | | | No | 34.91% |

## 4. Results

### 4.1. Reliability and Validity Analysis

In this paper, SPSS 22.0 was employed to assess the questionnaire's validity and reliability. The Kaiser–Meyer–Olkin (KMO) value was 0.734, and the reliability and validity were higher than 0.6. This proves that the variables involved in the questionnaire are highly correlated. Cronbach's alpha was equal to 0.736. When each variable in the model was tested for multicollinearity using STATA 16.0, the results revealed that none of the variables exhibited multicollinearity. The average variance inflation factor value was 1.79, with the highest variance inflation factor column value being 3.241. The likelihood ratio test result for the model was less than 0.01, indicating the model's overall relevance.

The validity and reliability of the three latent variables—economic expectations, social norms, and behavioral control—were investigated in the context of overall motivation. The specific outcomes are shown in Table 3. The Cronbach's alpha values of each latent variable ranged from 0.756 to 0.828, and all passed the consistency reliability test, indicating that the scale was reliable. The study model was then subjected to confirmatory factor analysis, the results of which are shown in the table below. The combined reliability is based on the CR value, and the CR value of each variable was between 0.700 and 0.905, and all were greater than the minimum threshold. Average variation extraction (AVE) is used to calculate discriminant validity, where an AVE greater than 0.5 indicates that the questionnaire has good discriminant validity. All three variables in this paper had AVE values greater than 0.5. The above indicators demonstrate that this questionnaire has good reliability and validity. The weight calculations of the factor analysis method were used to create scores for social norms and economic expectations, and the average value of perceived control and perceived difficulty was utilized to calculate the behavior control index of the business entities.

### 4.2. The Influence of Motivation on Willingness and Behavior Regarding Online Sales of Low-Carbon Agricultural Products

In accordance with theoretical analysis, this article applies STATA16.0 to model the behavior and online sales willingness of business entities. The results of the estimation obtained using each model are shown in Table 4. The P values for the LR statistics of the two models are significant, indicating that the model-related coefficients have high joint

importance. The two models are stable, with correct prediction rates of 75.47% and 81.13%, respectively.

**Table 3.** Latent random variable reliability and validity test results.

| Latent Random Variable | Measure Item | Factor Loading | AVE | CR | $\alpha$-Statistics |
|---|---|---|---|---|---|
| Economic Expectation | EE1 | 0.839 | 0.651 | 0.845 | 0.828 |
| | EE2 | 0.657 | | | |
| | EE3 | 0.884 | | | |
| Social Norms | SN1 | 0.626 | 0.598 | 0.813 | 0.804 |
| | SN2 | 0.846 | | | |
| | SN3 | 0.809 | | | |
| Behavioral Control | BC1 | 0.888 | 0.629 | 0.770 | 0.756 |
| | BC2 | 0.686 | | | |

**Table 4.** Estimation results for intention and behavior regarding online sales of low-carbon agricultural.

| Variable | Model 1 (Willingness) | | Model 2 (Behavior) | |
|---|---|---|---|---|
| | Coefficients | Standard Errors | Coefficients | Standard Errors |
| Economic Expectation | 0.577 ** | 0.240 | 0.850 *** | 0.255 |
| Social Norms | 0.355 * | 0.210 | 0.456 ** | 0.231 |
| Behavioral Control | −0.559 ** | 0.232 | −0.215 | 0.223 |
| Business entity scale | 0.726 *** | 0.277 | 0.286 | 0.279 |
| Sex | −0.198 | 0.306 | −0.597 * | 0.313 |
| Age | −0.183 | 0.190 | 0.169 | 0.193 |
| Proportion of non-agricultural income | 0.306 ** | 0.148 | 0.253 * | 0.150 |
| Constant term | −2.516 ** | 1.068 | −5.883 *** | 1.342 |
| Number of obs | 106 | | 106 | |
| Log likelihood | −48.249 | | −46.381 | |
| Pseudo R2 | 0.2568 | | 0.3686 | |
| LR chi2 | 33.34 *** | | 54.15 *** | |

***, **, * indicate significance at levels of 1%, 5% and 10%, respectively.

According to the regression results of models 1 and 2, economic expectations favorably affect business entities' willingness and behavior to engage in online sales at the 5% and 1% significance levels, respectively. This shows that business entities with a more favorable attitude towards online sales model are more likely to anticipate that doing business online will boost their income, sales volume and customer communication. When they predict that their income will be higher, they are also more likely to be willing to develop their online sales channels. The stronger an enterprise's presence, the more likely it is to make online sales. One reason for this may be that good economic expectations will support managers in making investments. Managers will therefore invest more time and effort into online sales. When an enterprise is stronger, it has a greater ability to conduct new types of sale. As a result, H1a and H2a are regarded as proven, with social norms having a direct influence on business entities' willingness and behavior. Social norms have a positive and significant impact on behavior and willingness, with effects exceeding the 10% and 5% significance levels, respectively. It stands to reason that strong bonds based on family, kinship, and business ties exist in rural communities. When considering whether to engage in online sales, business entities consider the opinions of family members, friends, agricultural technicians, other business entities, and the public. Their proclivity to sell online is greater, and they are more likely to adopt behaviors as they become better acquainted with online sales platforms. As a result, H1b and H2b are regarded as proven.

Behavioral control is presumed to have a direct effect on online sales willingness and behavior. The behavioral control variables failed the behavioral model test because they only met the willingness model's negative significance level of 5%. This indicates that the simplicity with which operators are able to understand online sales will boost their preparedness to adapt, lending credence to H1c, which is regarded as proven. Because of the disparity between business entities' perception of online sales prior to participation and their actual feelings regarding online sales following participation, operators may overestimate their level of engagement in online sales in terms of specific implementation behaviors. As a result, H2c is deemed to be disproven.

Furthermore, Models 1 and 2 show that the greater the magnitude of the control variables for business entities, the greater their tendency to sell online. In the willingness and behavior models, the percentage of an operator's income obtained from non-agricultural sources fulfilled positive significance tests of 5% and 10%, respectively, suggesting that, with increasing non-agricultural income, concurrent employment levels and time spent on agriculture also increase. This increases the amount of labor required to create online sales channels, preventing operators from introducing new sales channels.

### 4.3. Moderating Effects on the Transformation of Willingness to Sell Low-Carbon Agricultural Products Online into Behavior

The results of the test for the moderating effect of ability on the transformation from willingness to behavior with respect to online sales of low-carbon agricultural products are shown in Table 5. According to Models 3-1 and 3-2, natural conditions and the interaction variables between natural conditions and online sales willingness do not pass the significance test. The results show that improvements in natural conditions do not promote the conversion of willingness into behavior. One reason for this may be that the land transfer rate has reached a high level. The land management status of agricultural business operators is positive. As a result of ecological development, the city's total sown crop area has declined rapidly, and the rate of land turnover is approaching 90%. According to the results of this survey, 83.0% of agricultural business entities have more than 6.67 hectares of arable land, and more than 60% of their land is in a contiguous operation state. Those in charge are more concerned with improving other aspects than with improving natural conditions. In models 4-1 and 4-2, the policy conditions and the interaction variables between policy conditions and online sales willingness passed the 1% positive test level. This shows that policy conditions can facilitate the transformation of willingness into behavior. One reason for this may be that policies can provide subsidies in areas such as finance, taxation, and marketing. These efforts can encourage firms to take the lead in online sales and increase their understanding of and participation in online sales. These efforts increase potential benefits while reducing input costs. In other words, when external policy conditions are favorable, business entities are more likely to engage in online sales behavior. H3 assumes that increasing policy conditions in the opportunity dimension will help to increase the driving force of online sales willingness.

The results of the test to determine the moderating effect of ability on the transformation from willingness to behavior of online sales of low-carbon agricultural products are shown in Table 6. Factor analysis was performed on four indicators: the level of education of the operator, non-agricultural income, online sales operations expertise, and the frequency of internet surfing for agricultural product information. The cumulative variance explanation rate after rotation is 69.07%, indicating that the study data are suitable for factor analysis, and that the total score for the ability factors was acquired. The regression coefficients for the online sales intention and behavior were different between the high-ability group and the low-ability group, as evidenced by the findings using Models 6-1 and 6-2, with each F-statistic passing the test. The ability component clearly moderates the link between online sales intention and online sales behavior, illustrating how easily high-ability business entities can translate their online sales intention into online sales activity. One reason for this may be that more educated business entities prefer new ways of selling. They

are more receptive to new things. Business entities with higher internet sales operation ability are more skilled at conducting internet sales. In addition, frequent browsing for information on agricultural products conducive to developing a timely understanding of the market. All of these are conducive to the online sale of low-carbon agricultural products.

**Table 5.** The moderating effect of opportunity on the relationship between willingness and behavior.

| Variable | Model 3-1 | | Model 3-2 | | Model 4-1 | | Model 4-2 | |
|---|---|---|---|---|---|---|---|---|
| | Coefficients | Standard Errors | Coefficients | Standard Errors | Coefficients | Standard Errors | Coefficients | Standard Errors |
| Willingness | 0.725 *** | 0.085 | 0.719 *** | 0.086 | 0.710 *** | 0.080 | 0.699 *** | 0.079 |
| Natural conditions | −0.347 | 0.052 | −0.040 | 0.053 | | | | |
| Willingness × natural conditions | | | −0.025 | 0.111 | | | | |
| Policy conditions | | | | | 0.095 *** | 0.038 | 0.091 *** | 0.037 |
| Willingness × Policy conditions | | | | | | | 0.155 *** | 0.085 |
| Control variable | control | | control | | control | | control | |
| Number of obs | | | 106 | | | | 106 | |
| R2 | 0.5031 | | 0.5033 | | 0.5300 | | 0.5453 | |
| Adjusted R2 | 0.4729 | | 0.4678 | | 0.5015 | | 0.5128 | |
| F-statistic | 16.70 *** | | 14.19 *** | | 18.60 *** | | 16.79 *** | |

*** indicate significance at the 1% levels, respectively.

**Table 6.** The moderating effect of ability on the relationship between willingness and behavior.

| Variable | Explained Variable: Behavior in Selling Low-Carbon Agricultural Products Online | | | | | |
|---|---|---|---|---|---|---|
| | Full Sample (Model 5) | | High Ability (Model 6-1) | | Low Ability (Model 6-2) | |
| | Coefficients | Standard Errors | Coefficients | Standard Errors | Coefficients | Standard Errors |
| willingness | 0.709 *** | 0.082 | 0.722 *** | 0.139 | 0.718 *** | 0.113 |
| Control variable | control | | control | | control | |
| Number of obs | 106 | 56 | 50 | 106 | 56 | 50 |
| R2 | 0.5002 | 0.4141 | 0.6072 | 0.5002 | 0.4141 | 0.6072 |
| Adjusted R2 | 0.4752 | 0.3555 | 0.5625 | 0.4752 | 0.3555 | 0.5625 |
| F-statistic | 20.02 *** | 7.07 *** | 13.60 *** | 20.02 *** | 7.07 *** | 13.60 *** |

*** indicates significant at level of 1% respectively.

## 5. Discussion

### 5.1. Why Does High Willingness Coexist with Low Behavior

The reasons for the low participation rate in online sales of low-carbon agricultural products by urban agricultural entities may include the following: First, urban agricultural entities do not have a strong awareness of low-carbon agricultural products: their declaration process is complicated, and entities do not have much time to devote to the declaration. Second, the foundations are weak: there is a lack of modern agricultural industry planning, and the capital investment guarantee mechanisms and supporting policies and measures are not perfect [35]. Third, professional teams are not perfect: there is a lack of professional development certification for technical teams, and a lack of relevant technical guidance for the certification of low-carbon agricultural products. Fourth, high quality is difficult to obtain, and the price is high. Consumers lack an understanding of low-carbon agricultural products, emphasizing inexpensive-but-acceptable products during the purchase process. The input cost of low-carbon agricultural products during the production process is much higher than that of ordinary agricultural products, resulting in prices that are significantly higher than those of ordinary agricultural products. This will seriously diminish farmers' enthusiasm, and it is impossible to highlight a price advantage of the product compared to the competition on the market. Fifth, government policies favoring low-carbon agricultural

products are not yet in place [36,37]. Favorable policy interventions will promote the transformation of willingness into behavior. Various subsidy policies could reduce production costs, meaning that more farmers and enterprises may be willing to produce low-carbon agricultural products. Sixth, problems among the enterprises themselves will also affect the conversion of willingness. Some businesses with lower levels of education have a lower degree of understanding of emerging marketing methods. Some operators are not good at using the Internet for sales, which also does not help in the conversion of willingness into behavior.

*5.2. The Following Policy Recommendations Are Proposed to Increase Online Sales Participation*

5.2.1. Deepen Urban Agricultural Business Entities' Awareness Regarding the Online Sale of Low-Carbon Agricultural Products

At present, the introduction of the low carbon label in China is in its initial stages, and consumers are relatively unfamiliar with low-carbon agricultural products. Only when consumers are guided towards the consumption of low-carbon agricultural products will the production end of new agricultural business entities take the initiative to use carbon labels and actively carry out "three products and one standard" certification. Therefore, the government and the agricultural sector should play leading and supporting roles. The government should strengthen publicity and the guidance of public opinion and give play to the exemplary role of public institutions. Traditional information channels such as periodicals and TV and emerging social platforms such as WeChat, Weibo and live streaming can be used to increase public knowledge related to carbon labels and the implementation of the carbon label system, thus cultivating an overall awareness of low-carbon consumption and promoting low-carbon consumption behavior.

First, in terms of offline publicity, village committees can put up slogans and banners promoting low-carbon agricultural products in important public places in villages. Village committees can also issue leaflets and guidelines for the online sale of low-carbon agricultural products. In terms of online publicity, village committees can use the official WeChat public account or short video platforms such as Douyin and Kuaishou to produce popular science videos and short videos. This will promote low-carbon agricultural products and online sales, and strengthen the new agricultural business entities' understanding of low-carbon agricultural products and online sales. Secondly, a good online sales atmosphere should be created. This should make full use of the advantage of trust among local acquaintances in order to carry out learning activities among agricultural operating entities that have reached a certain level of achievement in terms of the online sale of low-carbon agricultural products. A model should be set up, enhancing the role of social demonstration and enhancing the awareness of operating entities regarding the online sale of low-carbon agricultural products. Finally, village committees should use their spare time to carry out online sales training or question-and-answer activities that are novel in form and rich in content related to low-carbon agricultural products. This will help business entities to master the operational skills of online sales. In terms of online training, each village or township can establish WeChat or QQ communication groups for online sales. Agricultural technology extension specialists can be arranged to answer questions and solve problems related to the online sales process of new agricultural business entities in a timely manner.

5.2.2. Create External Opportunities for the Development of Online Sales of Low-Carbon Agricultural Products

First, the government needs to develop innovative subsidy methods for online sales and strengthen policy support. In terms of taxation, new agricultural operators who plant low-carbon agricultural products should be given tax incentives, to a certain extent. With respect to policy related to the online sale of low-carbon agricultural products, the replacement of subsidies with awards should be considered, and special marketing subsidies should be set up, along with other tools to help agricultural operators reduce the financial pressure of online sales. Secondly, the government should take the lead in regulating the market order of carbon-label products in order to enhance consumer trust in carbon labels.

In terms of carbon label certification bodies, access thresholds should be established, and relevant standards for carbon label certification and issuance should be set, and neutral third-party institutions with high social prestige should be established to calculate the quantitative results for carbon labels, review the results and issue the certificates. Finally, a carbon label product credit system and an information disclosure platform should be established under public supervision, thus enhancing the sharing and transparency of carbon label information.

### 5.2.3. Identify Differences in the Abilities of Business Entities and Provide Accurate Services

Stable economic income and higher levels of access to Internet technology are the premise of and guarantee for the development of online sales channels for low-carbon agricultural products among business entities. When implementing incentive policies, the government should consider the heterogeneity of ability among management subjects. In high-ability groups, the government should stimulate their radiation-driven effect. For low-ability groups, the government should encourage the expansion of online sales channels for low-carbon agricultural products with other business entities.

### *5.3. Theoretical Contributions and Research Limitations*

This paper makes two marginal contributions. Firstly, in this paper, the theory of planned behavior was integrated with the theory of MOA, and the subjective perception and objective constraints faced by new urban agricultural operators when selling low-carbon agricultural products online were discussed, and the generation of online sales intention and online sales behavior among new agricultural operators specifically analyzed. This represents an innovation in terms of both research perspective and research methods. Secondly, in this paper, the formation of online sales willingness and behavior was revealed among new agricultural management subjects, and the mechanism of the transformation of online sales willingness into behavior was discussed.

There are a few shortcomings in this paper. On the one hand, we only analyzed the online sales of new agricultural entities in the research area. In addition, the producers of low-carbon agricultural products also included small farmers. The overall sales process of low-carbon agricultural products cannot be separated from their production. Small farmers' acceptance of online sales, their level of education, and other factors will have an impact on the overall outcomes. Ways of promoting the integration of these small farmers into online sales need to be further explored. On the other hand, few variables were included in our research framework, so the consideration of the problems may not have been comprehensive enough, and the influence of other factors on the online sales intention and behavior of new agricultural business entities was ignored, resulting in the research conclusions not being sufficiently comprehensive. Finally, although we strictly controlled the questionnaire process, the data in this paper may be subject to some chance. The data produced on the basis of the 106 questionnaires may have some data bias. The number of questionnaires was too small to ensure the accuracy of the study. This is something that needs to be improved upon in our future research.

### 6. Conclusions

In this paper, the theory of planned behavior and the MOA model were combined to investigate the factors that affect online sales willingness and behavior transformation among business entities. The following was discovered:

First, economic expectations and social norms can significantly increase willingness to sell online. Behavioral controls have the opposite effect during the formulation stage. The mechanism of action is as follows: Agricultural business entities expect to make a lot of money from online sales when there are examples of successful online sales nearby. In addition, they would be more eager to sell online if it were easier to overcome the technical challenges of online sales.

Second, behavioral control alone had no discernible impact on online sales behavior during the response stage among the motivational components. This could be due to a mismatch between the business entities' perception before participating in online sales and their actual experience afterward.

Third, the elements of opportunity and ability were included to close the gap between online sales willingness and behavior. For opportunity, natural conditions are not one of the possible factors influencing how online sales intention becomes action. The most likely explanation for this is that commercial entities place a greater emphasis on improving individuals' personal and professional abilities than on improving the environment. When government departments aggressively promote online sales and offer factor subsidies, policy settings can encourage the transition from online sales willingness to behavior. Among the ability factors, the higher the ability of a business entity, the better it will be able to realize the transformation from online sales willingness to behavior.

**Author Contributions:** Conceptualization, P.X.; methodology, F.X. (Fan Xu); software, M.L. and F.X. (Fan Xu); validation, F.X. (Fangke Xu) and F.X. (Fan Xu); formal analysis, F.X. (Fan Xu); investigation, F.X. (Fan Xu); resources, M.L., P.X. and F.X. (Fangke Xu); data curation, Y.L.; writing—original draft preparation, P.X., F.X. (Fan Xu) and F.X. (Fangke Xu); writing—review and editing, F.X. (Fan Xu), F.X. (Fangke Xu) and Y.L.; visualization, F.X. (Fan Xu); supervision, M.L. and Y.L.; project administration, P.X., M.L. All authors have read and agreed to the published version of the manuscript.

**Funding:** This research was funded by Guangxi University Young and Middle-Aged Teachers Research Basic Ability Improvement Project, grant number 22KY0476; and Shanghai Philosophy and Social Science Planning Project, grant number 2018BGL015.

**Institutional Review Board Statement:** Not applicable.

**Informed Consent Statement:** Not applicable.

**Data Availability Statement:** Data generated or analyzed during this study are available from the corresponding author upon request.

**Acknowledgments:** We would like to express our gratitude to all those who helped us while writing this article.

**Conflicts of Interest:** The authors declare no conflict of interest.

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
