# Peer review of "Research on Online Sales of Low-Carbon Agricultural Products by New Urban Agricultural Business Entities: Evidence from Shanghai, China"

_sustainability, doi:10.3390/su151813477_

Round 1
Reviewer 1 Report
The article titled "Research on online Sales of low-carbon agricultural products by urban new agricultural business entities Evidence from Shanghai, China" presents interesting information on online sales of low-carbon agricultural products. However, there are some aspects that weaken the manuscript. Below I detail my comments:
The abstract does not show the paper's methodology in detail, as well as the main research findings. I believe that the abstract of the manuscript should be reformulated.
Some ideas from the introduction need to be rearranged in the methodology. For example, lines 71-72.
The authors must cite the references well in the main text, after each citation the reference number must be accompanied.
Although the objective of the research can be intuited, it is necessary that the authors clearly establish it in the introduction, lines 159-166.
The authors should review the hypotheses, they are not well numbered. Also, what did they mean by the H1b hypothesis? (social demonstration function).
One of the aspects that generates confusion in the manuscript is about the regression models generated, this was not explained in the methodology of the manuscript. In addition, the model validation criteria should be described in this section. Section 4.1 (results) is part of the methodology.
In the discussion section, authors must analyze and discuss their findings with respect to other sources in the literature, something that is lacking in the current manuscript.
The English quality of the manuscript is good.
Author Response
Dear reviewers,
We feel great thanks for your professional review work on our article. As you are concerned, there are several problems that need to be addressed. According to your nice suggestions, we have made extensive corrections to our previous draft, the detailed corrections are listed below. And we revised the English of the full text.
Point 1: The abstract does not show the paper's methodology in detail, as well as the
main research findings. I believe that the abstract of the manuscript should be reformulated.
Response 1: We have overhauled the abstract. We have revised many grammatical incompatibilities and abbreviations. The specific revisions are listed below. “Traditional agricultural business entities have environmental problems such as high energy consumption and high pollution. To achieve the goal of carbon capping and carbon neutrality, the government should encourage urban agricultural operators to sell low-carbon agricultural products online. This plays an important role in smoothing the connection between production and marketing, achieving industrial prosperity and promoting low-carbon agricultural development. The paper explores the formation and behavioral transformation of online sales intentions by combining the theory of planned behavior, Motivation-Opportunity-Ability (MOA) model and binary Probit regression model using data from 106 questionnaires. The study found that economic expectations and social norms can significantly improve online sales intentions during the formation stage of online sales intentions. Behavior control is not conducive to improving online sales intention. In addition, we found a "gap" between the willingness and behavior of urban agricultural operators to sell online. This gap requires activation and adjustment of opportunity and capability factors in the behavior transformation phase. Finally, we find that the strengthening of policy conditions and management capacity can facilitate the transformation of urban agricultural operators' willingness to sell online into behavior. This paper provides opinions for the online sales of low-carbon agricultural products. While we continue to deepen urban agricultural operators' knowledge of online sales, we should also pay attention to creating external opportunities suitable for the development of online sales and identify the capacity differences among operators. This will provide precise services”
Point 2: Some ideas from the introduction need to be rearranged in the methodology. For example, lines 71-72.
Response 2: We have modified the repetition of views. We further analyzed Shanghai's transportation advantages and policy advantages. The specific changes are as follows. “This study chooses the basic components of agricultural management in Shanghai as the research object. Shanghai has a dense transportation network. Good transportation capacity ensures the supply of agricultural products. Shanghai has a policy advantage. Shanghai has introduced many agricultural support policies, which are conducive to the production and online sales of low-carbon agricultural products.”
Point 3: The authors must cite the references well in the main text, after each citation the reference number must be accompanied.
Response 3: We are very thankful to the reviewer for noticing the problem. We have made changes for the missing.
Point 4: Although the objective of the research can be intuited, it is necessary that the authors clearly establish it in the introduction, lines 159-166.
Response 4: We clarified the objectives of the paper in the introduction. And we revised the whole paragraph. The specific changes are as follows. “In summary, the transformation between the willingness and behavior of firms in urban areas for online sales of low-carbon agricultural products and its main influencing factors are unclear, and we attempt to fill this gap.” “The results of the study reveal the mechanism of the role of opportunity and capability factors in the path from willingness to behavior. And the findings provide constructive suggestions for the government to promote the development of low-carbon agriculture and online sales.”
Point 5: The authors should review the hypotheses, they are not well numbered. Also, what did they mean by the H1b hypothesis? (Social demonstration function).
Response 5: We have carefully revised the hypothesis of the article. We have added a note for hypothesis 2. Hypothesis 1 belongs to the assumption of willingness. Hypothesis 2 is a behavioral hypothesis. The social demonstration belongs to a writing error at the beginning. We have revised it to social norms. The specific revisions are as follows. “These norms usually refer to social norms. Online sales can be influenced by the people and environment around them.” “H2a: The greater the economic expectation of online sales, the more likely urban agricultural operators are to sell low-carbon agricultural products online.H2b: The stronger the social norms function of online sales, the more likely urban agricultural operators are to sell low-carbon agricultural products online.H2c: The easier it is to control online sales, the more likely urban agricultural operators are to sell low-carbon agricultural products online.”
Point 6: One of the aspects that generates confusion in the manuscript is about the regression models generated, this was not explained in the methodology of the manuscript. In addition, the model validation criteria should be described in this section. Section 4.1 (results) is part of the methodology.
Response 6: We relegated Section 4.1 to the methodology. And we specified the values of KMO test, Cronbach test, CR test and AVE test.
Point 7: In the discussion section, authors must analyze and discuss their findings with respect to other sources in the literature, something that is lacking in the current manuscript.
Response 7: We have discussed the results thoroughly. We have added an analysis of the regression results in both Section 4 and Section 5. It includes the analysis of factors such as natural conditions, government policies, and education level. The specific changes are as follows. “The results show that the improvement of natural conditions does not promote the con-version of willingness to behavior. The reason may be that the land transfer rate has reached a high level. The land management status of agricultural business operators is positive.” “This shows that policy conditions can facilitate the transformation of willingness into behavior. The reason for this may be that policies will provide subsidies in areas such as finance, taxation and marketing. These efforts can encourage firms to take the lead in online sales and increase their understanding of and participation in online sales. These efforts increase potential benefits while reducing input costs.” “The reason for this may be that the more educated business entities prefer new ways of selling. They are more receptive to new things. Business entities with higher internet sales operation ability will be more skilled in conducting internet sales. And frequent browsing of agricultural products information is more conducive to timely understanding of the market. All these are conducive to the low-carbon agricultural products online sales.” “Fifth, government policies favoring low-carbon agricultural products are not yet in place. Favorable policy interventions will promote the transformation of willingness to behavior. Various subsidy policies will reduce production costs, and more farmers and enterprises will be willing to produce low-carbon agricultural products. Sixth, the problems of enterprises themselves will also affect the conversion of willingness. Some businesses with lower education levels have a lower degree of understanding of emerging marketing methods. Some operators are not good at using the Internet for sales also do not help the conversion of willingness.”
We greatly appreciate the time and effort spent by the reviewers.
Thank you and best wishes.
Yours sincerely,
Fan Xu
June 9, 2023
Reviewer 2 Report
The problem of carbon emitted from agricultural activities is a current topic all over the world. Your approach is coherent and touches on topical issues. Congratulations for this data systematization.Author Response
Dear reviewers,
We greatly appreciate your time and efforts in the review process.
Thank you and best wishes.
Yours sincerely,
Fan Xu
June 9, 2023
Reviewer 3 Report
The authors integrated the theory of planned behavior with the theory of motivation-opportunity-ability (MOA) to analyze the subjective perception and objective constraints faced by urban new agricultural business entities in Shanghai, China when selling low-carbon agricultural products online. In general, the article is well organized in a logical way and the results were presented with meaningful explanation and discussion. However, this article has some minor drawbacks as follows:
- There are several places where the abbreviation was not expanded in full terms for the first time: e.g. Line 13 “MOA”; Line 201 “TPB”; Line 384 “SPSS”; Line 385 “KMO” etc.
- From Line 67 to 69, please explain why low-carbon agricultural products with higher prices have price advantages over ordinary agricultural products.
- Formatting issue: misalignment in Table 2. Also, recommend changing the table styles for all tables by adding more border lines for a clearer layout.
- From Line 575 to 578. The first research limitation was not clearly explained. Please add more context here.
Based on the above-mentioned comments, the reviewer believes this article might be suitable for publishing in Sustainability after revisions. Thank you for your submission.
Overall English is fine, minor editing of the English language is required. Please have someone with more proficient language skills review the article to smooth some of the expressions and correct minor grammar issues throughout the article. Thank you.
Author Response
Dear reviewers,
We feel great thanks for your professional review work on our article. As you are concerned, there are several problems that need to be addressed. According to your nice suggestions, we have made extensive corrections to our previous draft, the detailed corrections are listed below. And we revised the English of the full text.
Point 1: There are several places where the abbreviation was not expanded in full terms for the first time: e.g. Line 13 “MOA”; Line 201 “TPB”; Line 384 “SPSS”; Line 385 “KMO” etc.
Response 1: We have corrected the abbreviation. We have changed the abbreviation to Motivation-Opportunity-Ability,Theory of Planned Behavior,Kaiser-Meyer-Olkin. However, we did not modify SPSS because it is a very common statistical software. It would be very unnatural not to use abbreviations.
Point 2: From Line 67 to 69, please explain why low-carbon agricultural products with higher prices have price advantages over ordinary agricultural products.
Response 2: We are very sorry for this error. In fact the regular produce would have been more price advantageous. We have revised the error. “The price of low-priced agricultural products is usually higher than that of ordinary agricultural products. Ordinary agricultural products have obvious price advantages and premium advertising advantages.”
Point 3: Formatting issue: misalignment in Table 2. Also, recommend changing the table styles for all tables by adding more border lines for a clearer layout.
Response 3: We have revised the formatting and added border lines to Tables 3 and 4.
Point 4: From Line 575 to 578. The first research limitation was not clearly explained. Please add more context here.
Response 4: We provide a more detailed discussion of the first limitation. The specific changes are as follows. “On the one hand, we only analyzed the online sales of new agricultural subjects in the research object. And the production subjects of low-carbon agricultural products also include small farmers. The overall sales process of low-carbon agricultural products cannot be separated from production. Small farmers' acceptance of online sales, their education level and other factors will have an impact on the overall. How to promote the integration of these small farmers into online sales needs to be further explored.”
We greatly appreciate the time and effort spent by the reviewers.
Thank you and best wishes.
Yours sincerely,
Fan Xu
June 9, 2023
Reviewer 4 Report
The manuscript in question is not the only study anyone has made. It is just a use of a new model for a limited operational unit. The data is not randomized and a limited number of survey outputs has been presented which fails to justify the merit for publication. The manuscript also contains 19% similarity which means that every 5th word is plagiarized. Still, I suggest a thorough review, reshaping, and analysis for a better presentation in the revised version.
Major revision is suggested.
ABSTRACT: As a general comment the abstract should stand alone and be representative enough to understand every section of the manuscript. I am afraid to say this is not provided here and the abstract does not stand alone. Most of the lines look like being translated from another language and do not provide any meaningful inference.
LINE 9-12: The sentence is too long and non-coherent. This should be rewritten. Also, the abstract should provide a little introduction and later it should reflect the problem and drawbacks of current practices utilized by agricultural business entities (ABEs).
LINE 14-20:The results have been non-numerically explained and no information about experimental design has been provided in the abstract. How can you just write something is “significant” without any justification?
INTRODUCTION: Generally, the introduction section supports and explains the topic selection but here the introduction chapter is unfortunately not linked to title of the manuscript. Conclusively, the introduction is “excessively unrelated information”. As I have mentioned above that this document looks translated and poor English is observed. Here are a few examples;
LINE 43:*tight time*
LINE 44-47:Very long and casual sentencing.
LINE 47-50:Which committee? Where is a citation? Where is the supplementary material? I do not think the authors have read this version carefully.
LINE 50-56:This is not your work and your own thought. This has been established previously, this needs to be cited.
LINE 71-85:Why you suddenly start introducing your work in the middle of the introduction which technically is the last part?
METHODOLOGY: Questionnaire-based study and data collection via interview are not easy to publish. When the researchers use this method extensive data and vigorous statistical analysis are presented. Here I can only find a correlation study which is not enough to justify the merit. Even Shanghai cannot be represented with 106 retrieved questionnaires. There are some serious concerns over the model selection, randomization, data quantity, and also the analysis.
Language needs a lot of improvement. I can't mention everthing but it all looks translated or pyrated somewhere.
Author Response
Dear reviewers,
We feel great thanks for your professional review work on our article. As you are concerned, there are several problems that need to be addressed. According to your nice suggestions, we have made extensive corrections to our previous draft, the detailed corrections are listed below. And we revised the English of the full text.
Point 1: 1.ABSTRACT: As a general comment the abstract should stand alone and be representative enough to understand every section of the manuscript. I am afraid to say this is not provided here and the abstract does not stand alone. Most of the lines look like being translated from another language and do not provide any meaningful inference. LINE 9-12: The sentence is too long and non-coherent. This should be rewritten. Also, the abstract should provide a little introduction and later it should reflect the problem and drawbacks of current practices utilized by agricultural business entities (ABEs). LINE 14-20:The results have been non-numerically explained and no information about experimental design has been provided in the abstract. How can you just write something is “significant” without any justification?
Response 1: We made careful revisions. We rephrased long sentences and added some sentences. The specific changes are as follows. Traditional agricultural business entities have environmental problems such as high energy consumption and high pollution. To achieve the goal of carbon capping and carbon neutrality, the government should encourage urban agricultural operators to sell low-carbon agricultural products online. This plays an important role in smoothing the connection between production and marketing, achieving industrial prosperity and promoting low-carbon agricultural development. The paper explores the formation and behavioral transformation of online sales intentions by combining the theory of planned behavior, Motivation-Opportunity-Ability (MOA) model and binary Probit regression model using data from 106 questionnaires. The study found that economic expectations and social norms can significantly improve online sales intentions during the formation stage of online sales intentions. Behavior control is not conducive to improving online sales intention. In addition, we found a "gap" between the willingness and behavior of urban agricultural operators to sell online. This gap requires activation and adjustment of opportunity and capability factors in the behavior transformation phase. Finally, we find that the strengthening of policy conditions and management capacity can facilitate the transformation of urban agricultural operators' willingness to sell online into behavior. This paper provides opinions for the online sales of low-carbon agricultural products. While we continue to deepen urban agricultural operators' knowledge of online sales, we should also pay attention to creating external opportunities suitable for the development of online sales and identify the capacity differences among operators. This will provide precise services.
Point 2: INTRODUCTION: Generally, the introduction section supports and explains the topic selection but here the introduction chapter is unfortunately not linked to title of the manuscript. Conclusively, the introduction is “excessively unrelated information”. As I have mentioned above that this document looks translated and poor English is observed. Here are a few examples.
Response 2: We are very grateful for the details mentioned by the reviewer. We've made all the changes to this. We pay more attention to the use of grammar. We updated the citation and also put the inappropriate paragraph at the end. The specific changes are as follows. “less time” “Low-carbon agricultural products also have the biological characteristics of being perishable and difficult to keep fresh [4]. Moreover, the production of agricultural products is regional and seasonal. This determines that farmers have less time and heavy tasks in the sales process. The rapid development of online sales can provide new sales channels for low-carbon agricultural products.” “The No. 1 document issued by the Central Committee of the Communist Party of China and the State Council” “However, the actual participation rate of online sales of low-carbon agricultural products is low in Shanghai due to factors such as the limited knowledge of e-commerce operators, the imperfection of the existing logistics system, and the high cost of developing new sales channels. Under the background of achieving the goal of "double carbon", it is very necessary for us to explore the factors that affect the willingness of Shanghai's new agricultural business entities to sell low-carbon agricultural products online. This is not only conducive to improving the development level of low-carbon agriculture. It can also provide consumers with high-quality low-carbon agricultural products, thereby promoting the development of low-carbon agriculture.”
Point 3: METHODOLOGY: Questionnaire-based study and data collection via interview are not easy to publish. When the researchers use this method extensive data and vigorous statistical analysis are presented. Here I can only find a correlation study which is not enough to justify the merit. Even Shanghai cannot be represented with 106 retrieved questionnaires. There are some serious concerns over the model selection, randomization, data quantity, and also the analysis.
Response 3: In terms of data, our research group and research team have conducted a lot of preliminary research. In the early stage, we spent several months to determine the business entity. The 15 business entities involved in the 106 questionnaires were not randomly selected by us. Our questionnaire is also very detailed. We ensure that the data collected are representative. We also made changes to the errors in the results. For example, we add explanations to the results, we discuss limitations and further test the hypotheses.
We greatly appreciate the time and effort spent by the reviewers.
Thank you and best wishes.
Yours sincerely,
Fan Xu
June 9, 2023
Round 2
Reviewer 1 Report
The authors have addressed my comments and I consider that the manuscript has improved significantly. However, in the discussion of the manuscript, no references are added to support the information on theoretical contributions or policy recommendations. The authors should add references that contribute to the discussion of their findings with respect to the literature.
Engish quality of the manuscript is good
Author Response
Dear reviewers,
We feel great thanks for your professional review work on our article. As you are concerned, there are several problems that need to be addressed. According to your nice suggestions, we have made extensive corrections to our previous draft, the detailed corrections are listed below.
# Reviewer 1
The authors have addressed my comments and I consider that the manuscript has improved significantly. However, in the discussion of the manuscript, no references are added to support the information on theoretical contributions or policy recommendations. The authors should add references that contribute to the discussion of their findings with respect to the literature.
Thank you very much for your comments. We have added three recent references in section 5.1 to support our views and findings. The references are from leading journals and their views are very convincing. Section 5.2 is a recommendation, so we have not added references. The specific literatures are as followed.
- Luo J, Hu M, Huang M, et al. How does innovation consortium promote low-carbon agricultural technology innovation: An evolutionary game analysis. Journal of Cleaner Production, 2023, 384: 135564.
- Fan P, Mishra A K, Feng S, et al. The impact of China’s new agricultural subsidy policy on grain crop acreage. Food Policy, 2023: 102472.
- Liu T, Xu H. Post-assessment in policy-based strategic environmental assessment: Taking China's agricultural support and protection subsidy policy as an example. Environmental Impact Assessment Review, 2023, 100: 107047.
We greatly appreciate the time and effort spent by the reviewers.
Yours sincerely,
Fan Xu